# The Interplay Between Obesity and Venous Thromboembolism: From Molecular Aspects to Clinical Issue

**DOI:** 10.3390/ijms262110292

**Published:** 2025-10-22

**Authors:** Patrycja Sandra Zawadzka, Anna M. Imiela, Piotr Pruszczyk

**Affiliations:** Department of Internal Diseases and Cardiology, Centre for Management of Venous Thromboembolic Disease, Medical University of Warsaw, 02-005 Warsaw, Poland; s090641@student.wum.edu.pl (P.S.Z.); piotr.pruszczyk@wum.edu.pl (P.P.)

**Keywords:** adipose tissue, obesity, cardiovascular diseases, venous thromboembolism, leptin

## Abstract

This review examines the intricate relationship between obesity and venous thromboembolism (VTE), highlighting the underlying pathophysiological mechanisms and clinical implications. Obesity is an established independent risk factor for VTE, which includes deep vein thrombosis (DVT) and pulmonary embolism (PE). The risk of VTE escalates with increasing body mass index (BMI) and is particularly associated with abdominal adiposity. Dysfunctional adipose tissue (AT) in obesity promotes a pro-thrombotic state through chronic low-grade inflammation and impaired fibrinolysis. This inflammation is driven by stress within hypertrophied adipocytes, which leads to localized hypoxia, cellular dysfunction, and ultimately, cell death. This inflammation is driven by adipocyte stress and the infiltration of immune cells. The adipokine leptin exemplifies the complex link between obesity and VTE. While leptin has pro-thrombotic effects, low leptin levels are paradoxically associated with an increased morbidity and mortality in patients with acute PE, a phenomenon termed the “obesity paradox”. Furthermore, metabolic syndrome significantly increases the risk of recurrent VTE, with the risk growing with each additional metabolic component. Ultimately, a deeper understanding of the molecular and cellular links between obesity and VTE is essential for developing targeted strategies to reduce risk and improve outcomes in this vulnerable population.

## 1. Introduction

Obesity has emerged as a major global health concern of the 21st century, recognized as a complex metabolic disorder with extensive systemic consequences [1]. Among its most severe complications is its established role as an independent risk factor for venous thromboembolism (VTE) [2]. VTE, which presents as either deep vein thrombosis (DVT) or pulmonary embolism (PE), ranks as the third most common acute cardiovascular syndrome, following myocardial infarction and stroke in frequency [3].

On the molecular level, thrombosis in obesity can arise through several pathways, including the effects of adipose tissue-derived cytokines like leptin and adiponectin, enhanced coagulation processes coupled with impaired fibrinolysis, persistent inflammation, oxidative stress leading to endothelial damage, and metabolic disturbances involving lipids and glucose commonly seen in metabolic syndrome (MetS) [4]. This dysregulation is particularly evident in the opposing actions of leptin and adiponectin. Leptin promotes a pro-thrombotic state by impairing fibrinolysis and enhancing platelet activation, an effect supported by genetic studies linking its high levels to a significantly increased VTE risk [5]. In contrast, adiponectin is considered a potential protective factor, and its reduced levels are associated with worse outcomes, such as post-thrombotic syndrome [5,6]. Furthermore, adiposity exacerbates platelet activity, leading to overly responsive platelets that are a core reason for the overall risk for thrombosis. This is demonstrated by the increased activation and aggregability observed in the platelets of obese individuals [7]. Clinically, patients with obesity are characterized by a higher amount of visceral fat, resulting in higher intra-abdominal pressure and blood stasis [8].

Obesity contributes to venous stasis through a clear mechanism known as obesity-induced caval vein compression syndrome [9]. The increased intra-abdominal pressure resulting from a high BMI and intra-abdominal fat accumulation is linked to a corresponding increase in the pressure differential observed across the diaphragm between the thoracic and abdominal vena cava segments. The resulting compression of the inferior vena cava hinders venous return, leading to reduced venous blood flow velocity and an increased risk for venous thrombosis [9].

Traditionally, the pathogenesis of VTE has been explained by Virchow’s triad: endothelial cell (EC) injury, stasis of blood flow, and a hypercoagulable state. More recently, a fourth factor—chronic inflammation—has gained attention, leading to the emerging concept of immunothrombosis [10]. In particular, visceral adipose tissue (VAT) appears to play a key role in promoting a systemic, chronic inflammatory environment, contributing to “adiposopathy” and enhancing the risk of thrombus formation [10]. Recent research pinpoints adipocyte death as a key driver of this inflammation, as their large size prevents efficient, anti-inflammatory removal by single macrophages [11]. Instead, this size issue triggers a pro-inflammatory response where multiple macrophages surround the dead cell to form characteristic crown-like structures (CLS). The primary cause of this inflammation is a functional change in the tissue’s existing macrophages, rather than the recruitment of new monocytes from the blood [11].

The intricate relationship between obesity and VTE is influenced by adipokines such as leptin, which exerts a paradoxical effect on thrombosis [12]. While leptin is generally associated with prothrombotic activity, clinical studies have demonstrated that lower leptin levels in patients with acute PE are predictive of increased morbidity and mortality [13]. This phenomenon, often referred to as the “obesity paradox”, is further supported by data from large prospective registries, which suggest that obese patients may experience lower mortality rates following a VTE event compared to their normal-weight counterparts [14,15].

However, while obesity may be associated with improved short-term survival after an initial thrombotic event, the broader constellation of metabolic abnormalities known as metabolic syndrome significantly increases the risk of recurrent VTE. Each additional metabolic component compounds this risk, directly linking poor metabolic health to long-term thrombotic vulnerability [16]. In addition, managing anticoagulation—a cornerstone of pharmacologic therapy for VTE—remains particularly challenging in patients with obesity (BMI ≥ 40 kg/m^2^), due to altered pharmacokinetics and pharmacodynamics, which can complicate clot resolution and therapeutic dosing.

This review critically examines the relationship between obesity and VTE, synthesizing current knowledge from molecular mechanisms to clinical outcomes. It places particular emphasis on the complex pathophysiology underlying this association, focusing on the systemic effects of obesity and the physiological changes that elevate cardiovascular and thrombotic risk. Drawing on scientific evidence, the review also explores how emerging findings can inform clinical practice, including risk assessment and treatment strategies. Ultimately, a deeper understanding of this connection is essential for developing targeted interventions that reduce complications and improve outcomes in this large and vulnerable patient population.

## 2. The Physiology of Adipose Tissue

### 2.1. Thermal Homeostasis

Adipose tissue (AT) plays a key role in regulating body temperature, functioning passively as an insulator to conserve body heat. Brown adipose tissue (BAT) acts as a major site of metabolic heat production [17]. The basic role in thermogenesis is played by mitochondria. Thermogenin, known as uncoupling protein 1, or UCP1, is a mitochondrial carrier protein found in BAT, and it generates heat by uncoupling adenosine triphosphate (ATP) synthesis from oxygen consumption in the process called “non-shivering thermogenesis”. It accomplishes this by creating a bypass for protons across the inner mitochondrial membrane, releasing its stored potential energy as heat rather than using it to produce ATP [18]. The activation of this thermogenic process is regulated by several signaling pathways, with the β3-adrenergic signaling pathway playing the primary role [19]. First, norepinephrine stimulates adenylyl cyclase, which in turn synthesizes cyclic AMP (cAMP). The increase in cAMP levels activates protein kinase A (PKA), which then promotes the transcription of UCP1 and other thermogenic factors [20]. In addition to the primary β-adrenergic pathway, other molecules contribute to the regulation of AT thermogenesis. The positive regulatory domain zinc finger region protein (PRDM16) is a critical transcription factor for establishing brown fat characteristics. PRDM16 activates peroxisome receptor γ coactivator 1α (PGC-1α), which coordinates mitochondrial biogenesis and upregulates the expression of UCP1 [21].

### 2.2. Energy Storage

AT is specialized for the efficient storage of excess metabolic energy in the form of triacylglycerols (TAGs), which are compartmentalized into intracellular organelles known as lipid droplets [22]. These droplets are characterized by a phospholipid monolayer that surrounds a hydrophobic core composed primarily of TAGs and sterol esters [23]. The function of lipid droplets is largely determined by their associated proteins, with the perilipin (PLIN) family playing a central regulatory role. PLINs control the recruitment and activity of key metabolic enzymes, thereby governing the rate of lipolysis [24]. TAG synthesis, or lipogenesis, is a multi-step enzymatic process that occurs at the endoplasmic reticulum (ER). During the absorptive state, adipocytes secrete lipoprotein lipase (LPL), an enzyme that hydrolyzes TAGs in circulating lipoproteins, enabling the uptake of free fatty acids (FFAs) into the cell. Simultaneously, adipocytes absorb glucose, which is used to generate a glycerol backbone. The FFAs are then esterified to this glycerol, forming new TAGs for storage [25,26,27]. Alternatively, TAGs can be synthesized via de novo lipogenesis, a process active during both feeding and fasting states. Here, fatty acids are synthesized from acetyl-CoA and subsequently esterified with glycerol to form TAGs [28]. Conversely, during periods of elevated energy demand—such as fasting or physical exertion—hormonal signals activate lipolysis, the breakdown of stored TAGs [29]. This process is initiated by catecholamines released by the sympathetic nervous system [30]. The nervous system triggers an intracellular signaling cascade that activates two key enzymes: adipose triglyceride lipase (ATGL) and hormone-sensitive lipase (HSL). The coordinated action of ATGL and HSL results in the hydrolysis of TAGs into glycerol and FFAs, which can then be mobilized to meet the body’s energy needs [24,31].

### 2.3. Endocrine Functions

#### 2.3.1. Leptin

According to the literature, leptin is a well-known pro-inflammatory adipokine. Individuals with obesity are characterized by higher circulating levels of this hormone when compared to lean persons. Notably, an elegant study by Mrozinska et al. found that lower adiponectin and elevated leptin levels measured three months after DVT—regardless of obesity status—were associated with an increased risk of post-thrombotic syndrome (PTS) [6]. Leptin is postulated to contribute to an elevated risk of VTE by impairing fibrinolysis and promoting platelet activation [32].

Leptin is a protein hormone secreted by AT, and its synthesis is regulated by the *obese* (*ob*) gene [33]. It was the first endocrine hormone identified as originating from adipocytes. Leptin’s primary role is to regulate energy balance by suppressing appetite and signaling satiety to the brain [33,34]. Its concentration is positively correlated with AT mass. However, in individuals with obesity, leptin levels are often elevated without corresponding physiological effects—an indication of leptin resistance. Furthermore, leptin’s effect on appetite regulation is saturable, reaching a plateau at high physiological concentrations. This saturation may support long-term weight loss in individuals with chronically low leptin levels, who remain more sensitive to its effects [34].

The leptin receptor (OB-R) comprises three main domains: an extracellular region that binds the hormone, a transmembrane domain, and an intracellular region responsible for signal transduction. This structure classifies OB-R as a member of the class I cytokine receptor family, specifically the interleukin-6 (IL-6) family [31,32]. Among its isoforms, the long form—OB-Rb—is considered the primary signaling receptor [33]. Upon leptin binding, OB-Rb activates Janus kinase 2 (JAK2), leading to receptor phosphorylation and activation of downstream pathways, including the proliferative ERK/c-Fos cascade and the regulatory STAT3/SOCS3 feedback loop [35,36].

Studies involving recombinant leptin administration in *ob*/*ob* deficient mice, which lack endogenous leptin, demonstrate significant reductions in food intake and body weight, primarily through decreased adiposity and improved hyperglycemia [37]. Even in wild-type mice with normal leptin levels, exogenous leptin reduces body fat, highlighting its role in regulating energy storage [38]. This action phosphorylates the leptin receptor, triggering a positive ERK/c-fos pathway and an inhibitory feedback loop for the STAT3/SOCS3 pathway [39].

Recent research shows that intranasal administration of leptin induces anorectic effects and weight loss in diet-induced obese rats, mirroring the responses seen in lean controls. This method also selectively reduces intra-abdominal and epididymal fat stores. These outcomes are associated with altered hypothalamic expression of neuropeptides involved in appetite regulation. Importantly, intranasal delivery appears to bypass central leptin resistance, a key obstacle in obesity treatment [40].

Leptin’s interaction with its receptor leads to inhibition of orexigenic neuropeptides such as neuropeptide Y and agouti-related peptide, while simultaneously enhancing the expression of anorexigenic peptides, including proopiomelanocortin and brain-derived neurotrophic factor [41,42,43].

Leptin also plays a critical role in glucose homeostasis, as demonstrated in models of obesity and type 2 diabetes (T2D), such as leptin-deficient (*ob*/*ob*) and leptin receptor-deficient (*db*/*db*) mice. These models exhibit insulin resistance due to impaired leptin signaling. However, the mechanisms differ: *ob*/*ob* mice develop hyperinsulinemia from excessive insulin production, whereas *db*/*db* mice display dysfunctional insulin secretion [44]. Remarkably, leptin administration reverses hyperinsulinemia; even low doses can reduce insulin levels by 41%, with higher doses normalizing levels to those seen in lean mice [45]. Leptin exerts these effects through both central and peripheral mechanisms, including direct action on pancreatic β-cells to regulate their mass, insulin gene expression, and insulin secretion [46,47,48].

#### 2.3.2. Adiponectin

Adiponectin is a protein hormone primarily secreted by AT and is present in high concentrations in the circulatory system [49]. While circulating adiponectin levels are generally inversely associated with overall adiposity—as measured by Body Mass Index (BMI)—evidence suggests this relationship is also site-specific. One study demonstrated a significant inverse correlation between adiponectin levels and abdominal visceral fat, but a positive correlation with thigh subcutaneous fat [50]. Additionally, a strong negative association between plasma adiponectin and BMI has been reported, with non-obese individuals exhibiting an average concentration of 19.3 µg/mL, compared to 13.2 µg/mL in obese individuals. This inverse relationship was further quantified: each 1 kg/m^2^ increase in BMI was associated with a 0.44 µg/mL reduction in adiponectin levels [51].

Adipocytes synthesize and secrete multiple forms of adiponectin. The basic structural unit is a low-molecular-weight (LMW) trimer, which can assemble into medium-molecular-weight (MMW) hexamers and high-molecular-weight (HMW) oligomers [52,53]. In human circulation, adiponectin is predominantly present in the MMW and HMW forms, with the HMW isoform showing the strongest association with improved insulin sensitivity and lower blood glucose levels [52]. Notably, the HMW form also plays a critical role in cardiovascular health. Studies have found reduced levels of HMW adiponectin in patients with atherosclerosis. This isoform exerts cardioprotective effects by preventing apoptosis, promoting muscle cell regeneration, and supporting revascularization—mechanisms largely mediated through AMP-activated protein kinase (AMPK) signaling [54].

Activation of AMPK is essential for the subsequent phosphorylation of protein kinase Akt, a process mediated by phosphatidylinositol 3-kinase (PI3K). This signaling cascade ultimately leads to the phosphorylation of endothelial nitric oxide synthase, a critical enzyme in the regulation of vascular function [55]. These mechanisms suggest that fluctuations in HMW adiponectin levels may play a central role in the onset and progression of vascular complications associated with obesity-related diseases [54].

Adiponectin plays a central role in enhancing insulin sensitivity, primarily through its regulatory effects on hepatic metabolism. It decreases hepatic glucose production while promoting fatty acid oxidation in the liver [56]. Evidence from animal models supports this function: mice lacking the adiponectin gene develop insulin resistance, highlighting the hormone’s critical role in glucose homeostasis. Furthermore, in mice with high-fat diet-induced insulin resistance, a two-week adiponectin treatment significantly improved insulin sensitivity and reduced ectopic lipid accumulation in both the liver and skeletal muscle [57].

Based on the recent findings, the relationship between adipokines and VTE is complex. The Mendelian randomization study identified a strong causal link between high circulating leptin levels and an increased risk of VTE. Specifically, it found that genetically predicted high leptin levels were associated with a 96% increase in the odds of VTE, a 152% increase for DVT, and a 126% increase for PE. This genetic study also suggested that adiponectin is a potential protective factor, associated with a 15% decreased risk of VTE and a 19% decreased risk for PE, although these associations were no longer statistically significant after applying a False Discovery Rate correction [5]. In contrast, the Multi-Ethnic Study of Atherosclerosis (MESA) presented different findings. In this observational study, the initial association between higher leptin levels and VTE became non-significant after adjusting for BMI, suggesting the relationship is confounded or mediated by overall adiposity. Furthermore, the MESA study did not identify a statistically significant link between adiponectin levels and the likelihood of developing VTE [32]. These contrary findings should be explained in larger studies.

### 2.4. Immune Functions

Functioning as a dynamic organ with significant immunometabolic influence, AT has emerged as a central focus in immunology. While research on obesity has deepened our understanding of immune cell populations within AT, it has also underscored their essential role in maintaining homeostasis and immune readiness under healthy physiological conditions [58,59].

Adipocytes possess an intrinsic capacity for immune interaction, mediated through a variety of receptors responsive to external signals [58]. Among the most notable are Toll-like receptors (TLRs), which act as pattern recognition receptors (PRRs), detecting pathogen-associated molecular patterns (PAMPs) and initiating pro-inflammatory responses [60]. This immunological responsiveness is present throughout the adipocyte life cycle—from mesenchymal stem cells to fully differentiated adipocytes. For example, immature mesenchymal stem cells within AT express TLRs 1–6 and TLR9, indicating early immune potential [60]. This capability is retained and expanded in more mature cells; both preadipocytes and mature adipocytes in mice express a broader range of functional TLRs (1–9). Upon stimulation, these cells secrete pro-inflammatory cytokines such as IL-6, firmly positioning adipocytes as active participants in the body’s innate immune defense [61,62].

Interestingly, the inflammatory response triggered by TLR4 activation in response to the bacterial component lipopolysaccharide (LPS) is more pronounced in preadipocytes than in mature adipocytes. In contrast, the inflammatory effects of fatty acids—particularly saturated fatty acids (SFAs)—are now understood to involve more complex mechanisms than direct TLR4 activation. Although TLR4 is essential for mediating the pro-inflammatory response to SFAs, current evidence suggests that SFAs are not direct agonists of the receptor. Instead, the process appears to be sequential: an initial priming signal, such as LPS, reprograms cellular metabolism, rendering the cell more responsive. This metabolic reprogramming allows a subsequent SFA stimulus to induce an inflammatory response [63,64].

The secretion of pro-inflammatory mediators—including IL-6 and chemokines such as CCL2, CCL5, and CCL11—in response to LPS stimulation is mediated through activation of the nuclear factor κB (NF-κB) signaling pathway. However, more recent studies indicate that the saturated fatty acid palmitate induces inflammation via distinct mechanisms. Rather than activating the NF-κB pathway, palmitate exerts its pro-inflammatory effects primarily through alternative signaling routes, notably by inducing ER stress [63,64,65].

Under healthy, lean conditions, AT maintains immune homeostasis through the balanced secretion of adipokines. A central molecule in this regulatory environment is adiponectin, a key anti-inflammatory hormone. Adiponectin exerts its effects by suppressing pro-inflammatory signaling pathways—most notably through inhibition of the NF-κB pathway—while simultaneously enhancing anti-inflammatory responses [66]. It downregulates the production and activity of pro-inflammatory cytokines such as tumor necrosis factor-α (TNF-α) and IL-6 and promotes an anti-inflammatory state by inducing the expression of cytokines like interleukin-10 (IL-10) [67]. At the cellular level, adiponectin further contributes to immune regulation by facilitating the polarization of macrophages toward the M2 phenotype—a profile associated with tissue repair and resolution of inflammation—primarily through its inhibitory effect on TNF-α [68].

Through these complex molecular mechanisms, AT functions as an active sensory and signaling organ, capable of detecting immune and metabolic threats and coordinating systemic responses. However, in the context of obesity, this delicate homeostatic balance is disrupted, leading to chronic, low-grade systemic inflammation [69].

This process is primarily mediated by the chemokine-driven recruitment of inflammatory cells, a crucial element in the progression of cardiovascular diseases (CVDs). The adipokine chemerin is a critical link in this process, connecting obesity directly to the inflammatory state that damages the vascular system [70]. According to a pivotal study, circulating chemerin concentrations are significantly elevated in obese individuals (BMI > 30 kg/m^2^) compared to lean subjects (BMI < 25 kg/m^2^), with mean values of 296.5 ng/mL and 222.7 ng/mL, respectively [71]. This elevation is crucial for CVD pathogenesis because chemerin acts as a potent chemoattractant for inflammatory macrophages. The accumulation of these cells in fat tissue creates a persistent, low-grade inflammatory state that can damage endothelial cells and facilitate the formation of atherosclerotic plaques, which are hallmarks of CVD [72].

## 3. Inflammation

Obesity is marked by the expansion of white adipose tissue (WAT), a process driven by two primary cellular mechanisms: hypertrophy (an increase in the size of existing adipocytes) and hyperplasia (an increase in adipocyte number). Among these, hypertrophy is considered the predominant contributor to the accumulation of fat mass [70,73,74].

A growing body of evidence supports a strong association between obesity and chronic, low-grade systemic inflammation. Notably, adipose tissue itself serves as a major source of this inflammation. Rather than being merely a storage site, adipose tissue is a complex active organ comprising not only adipocytes but also a diverse stromal vascular fraction (SVF). The SVF contains various immune cells, including macrophages, T cells, B cells, natural killer T (NKT) cells, natural killer (NK) cells, and neutrophils—all of which contribute to the inflammatory environment in obesity [75,76,77,78].

One key mediator of immune cell recruitment in this context is the adipokine chemerin. Chemerin functions as a chemoattractant, guiding dendritic cells (DCs), macrophages, and lymphocytes to sites of inflammation, thereby reinforcing the link between adipose tissue dysfunction and the inflammatory state characteristic of obesity [70,79].

It is now well established that immune cell infiltration and the resulting inflammation within AT in obesity are key contributors to both local tissue damage and the systemic metabolic dysfunction that characterizes the disease [80]. The pathological expansion of AT is closely associated with widespread metabolic disturbances, including insulin resistance, hypertension, and dyslipidemia [73,74,75]. As a result, a strong inverse relationship exists between excess adiposity and metabolic health, with obesity significantly increasing the risk of non-communicable diseases such as T2D and CVD (Figure 1) [81,82,83,84].

### 3.1. Triggers of Obesity-Induced Inflammation

The initiation of inflammation in AT is a multifaceted process, driven by the cumulative effect of numerous interrelated stressors. These triggers originate both intrinsically—from within adipocytes—and extrinsically, through systemic influences. Together, they elicit an immune response that establishes a self-reinforcing feedback loop, amplifying and sustaining the inflammatory state [85,86].

In the context of chronic positive energy balance, adipocytes undergo hypertrophy to accommodate excess lipid accumulation. However, when this expansion exceeds a critical physiological threshold without corresponding angiogenesis, it results in cellular stress—most notably hypoxia and ER stress. These conditions, in turn, contribute to adipocyte dysfunction, including impaired insulin signaling and the onset of metabolic disturbances such as insulin resistance [87]. The subsequent recruitment of pro-inflammatory M1 macrophages to these sites leads to their aggregation around necrotic adipocytes, resulting in the formation of characteristic crown-like structures [88,89].

The death of adipocytes leads to the release of damage-associated molecular patterns (DAMPs) [85]. These signals are recognized by macrophages, which respond by activating the NLR family pyrin domain containing 3 (NLRP3) inflammasome. This activation triggers the secretion of the pro-inflammatory cytokine interleukin-1β (IL-1β), thereby contributing to local inflammation [90].

In parallel, metabolic stress induced by obesity causes various adipose tissue cells to undergo cell cycle arrest. Although these senescent cells no longer proliferate, they remain metabolically active and develop a senescence-associated secretory phenotype (SASP). This phenotype is characterized by the sustained release of pro-inflammatory molecules, further amplifying local inflammation and disrupting tissue homeostasis [91,92].

In the early stages of obesity, an increase in saturated FFAs drives excessive oxygen consumption by adipocytes, resulting in localized hypoxia and the activation of hypoxia-inducible factor 1-alpha (HIF-1α) [86,93]. Under normoxic conditions, prolyl hydroxylase domain proteins (PHDs) continuously mediate the degradation of HIF-1α. However, during hypoxia, PHD activity is suppressed, preventing HIF-1α degradation and allowing its accumulation within cells [94]. Elevated HIF-1α levels contribute to adipose tissue inflammation and dysfunction by promoting the recruitment of pro-inflammatory M1 macrophages. Immunohistochemical analyses support this mechanism, demonstrating increased accumulation of F4/80+ macrophages in hypoxic regions of adipose tissue [93,95].

### 3.2. Macrophages

Adipose tissue macrophages (ATMs) play a central role in mediating inflammation in obesity, thereby contributing significantly to the development of metabolic disorders [96,97]. The proportion of macrophages within adipose tissue correlates strongly and positively with body weight. In lean mice and humans, macrophages constitute less than 10% of the adipose tissue cell population, whereas in obese individuals, this percentage increases to approximately 40%, and can exceed 50% in cases of severe obesity in mice [96]. Supporting this, a recent study demonstrated marked macrophage infiltration into the VAT of high-fat diet–fed mice, a finding further validated by reanalysis of human datasets showing a similar trend in obese subjects [97]. This substantial accumulation of macrophages, combined with a shift in their functional phenotype—often toward a pro-inflammatory M1 state—positions them as key contributors to both local tissue damage and systemic insulin resistance [98].

A surprising developmental link has been proposed between adipocytes and macrophages, suggesting that these two cell types are not only closely related but may also possess the capacity to transform into one another—a phenomenon with particular relevance to obesity-induced inflammation [99,100]. This potential plasticity highlights the intimate and dynamic relationship between AT and its immune environment [101].

Both adipocytes and macrophages share common surface markers, such as F4/80, further supporting their developmental proximity. Moreover, peroxisome proliferator-activated receptor gamma (PPARγ) functions as a key regulatory factor in the differentiation and activity of both cell types. Notably, the suppression of PPARγ by microRNAs such as miR-27a has been shown to promote a shift toward a pro-inflammatory phenotype, contributing to the immune dysregulation observed in obesity [102].

Macrophage infiltration into adipose tissue during obesity arises from both the recruitment of circulating monocytes and the local proliferation of resident macrophages [75,103,104]. The recruitment process is primarily driven by chemoattractants secreted by stressed adipocytes, with the monocyte chemoattractant protein-1 (MCP-1/CCL2) pathway—acting through the CCR2 receptor—being the most well-characterized [105].

More recently, a novel pathway involving fibroblast activation protein (FAP) has been identified. FAP modulates the chemokine CCL8, forming a critical axis for monocyte recruitment. Disruption of the FAP–CCL8 interaction has been shown to reduce monocyte infiltration and confer protection against diet-induced obesity and inflammation [106].

Once recruited, ATMs can further expand their population through local proliferation, particularly in CLS that form around necrotic adipocytes [88,89]. In addition, the expression of inflammatory and metabolic factors—including IL-6, MCP-1, lipase, resistin, and leptin—is upregulated in obese AT. These factors enhance the expression of adhesion molecules on vascular endothelial cells, facilitating monocyte adhesion and transmigration into adipose tissue, thereby promoting ATM accumulation [107].

This increase in macrophage population is accompanied by a functional shift. In lean conditions, ATMs predominantly exhibit an anti-inflammatory M2 phenotype. However, in obesity, they transition to a pro-inflammatory M1-like state, marked by increased expression of CD11c and the production of type 1 cytokines such as TNF-α and IL-6 [102,108,109]. Supporting this, studies have shown that the frequency of CD11c^+^ monocytes is significantly higher in obese individuals compared to lean controls, although this proportion decreases following adherence to a low-fat diet [110].

The pro-inflammatory M1 macrophage phenotype is primarily maintained through signaling pathways activated by TLRs and interferon-γ (IFN-γ) [111,112]. IFN-γ, upon binding to its receptor (IFNGR), initiates a signaling cascade involving the activation of Janus kinases (JAKs), which subsequently phosphorylate and activate the downstream transcription factor signal transducer and activator of transcription 1 (STAT1) [112].

Concurrently, the recognition of PAMPs or DAMPs via TLRs promotes M1 polarization. TLR engagement activates a signaling cascade that culminates in the activation of the NF-κB pathway [113]. These converging signaling mechanisms drive the expression of a broad array of pro-inflammatory mediators, including cytokines such as TNF-α, IL-6, and IFN-γ, as well as chemokines like CXCL10 and CXCL11 [114]. Collectively, these secreted factors create a highly inflammatory microenvironment, establishing a positive feedback loop that both amplifies the initial immune response and reinforces the M1 macrophage phenotype [112,115,116,117].

## 4. Leptin and Clinical Implications

Obesity is a well-established risk factor for venous thromboembolism, promoting a pro-thrombotic state through sustained inflammation and impaired fibrinolysis [2]. One key contributor to this dysfunction is the elevated level of plasminogen activator inhibitor-1 (PAI-1) found in individuals with obesity, which suppresses fibrin clot degradation and thereby promotes thrombosis [118].

Leptin, an adipokine secreted by adipose tissue, has emerged as an independent cardiovascular risk factor due to its pro-thrombotic effects in both arterial and venous systems [119]. Mechanistically, leptin contributes to vascular pathology by stimulating platelet aggregation, facilitating arterial thrombus formation, and inducing oxidative stress within the vascular endothelium [119,120,121,122].

Interestingly, leptin’s role in VTE appears complex and somewhat paradoxical. While its pro-thrombotic properties are well-documented in the context of obesity, clinical data also indicate that low leptin levels may be associated with poorer outcomes in patients with acute pulmonary embolism (APE) [12].

In a study involving 264 patients with APE, lower baseline leptin levels were independently associated with worse prognosis. Patients who experienced a complicated 30-day clinical course had significantly lower median leptin levels (5.3 ng/mL) compared to those with an uncomplicated course (10.4 ng/mL). Similarly, lower leptin levels predicted higher long-term mortality, with non-survivors exhibiting a median level of 7.8 ng/mL versus 10.4 ng/mL in survivors. Overall, patients with the lowest leptin levels were 2.8 times more likely to experience complications within 30 days and had reduced long-term survival [13].

The unexpected association between low leptin levels and worse outcomes in patients with pulmonary embolism may be explained by the concept of the “obesity paradox”—a phenomenon suggesting that leptin’s role extends beyond merely reflecting body mass. In this context, leptin appears to exert protective effects, but these benefits are contingent on the individual’s sensitivity to the hormone [12]. This protective capacity is diminished in individuals with leptin resistance, a common metabolic disturbance in obesity [123].

One proposed mechanism for leptin resistance involves matrix metalloproteinase-2 (MMP-2), a proteolytic enzyme that becomes activated in the hypothalamus during obesity. MMP-2 cleaves the extracellular domain of the leptin receptor, thereby disrupting leptin signaling and contributing to the development of resistance [124]. This mechanism is supported by clinical evidence: in patient cohorts, high leptin levels were associated with a reduced risk of recurrent VTE—but only in individuals who demonstrated signs of leptin sensitivity, such as lower BMI and decreased MMP-2 levels [12]. In leptin-responsive individuals, the hormone may exert protective effects through multiple mechanisms. Leptin is known to modulate immune and inflammatory pathways, both of which are closely linked to the pathophysiology of PE [125,126]. Additionally, leptin may directly support vascular health by stimulating nitric oxide production and enhancing the function of endothelial cells and other components involved in vascular repair and regeneration (Table 1) [127].

## 5. The Association Between Obesity and VTE in Clinical Trials

Multiple studies have confirmed that obesity is a significant and independent risk factor for venous thromboembolism [2,12,128,132,134]. Data from a case–control study indicate that obese individuals are 6.2 times more likely to develop VTE compared to those with normal weight. Moreover, a clear dose–response relationship exists between the severity of obesity and VTE risk: the odds ratio increases from 2.857 for class I obesity, to 5.293 for class II, and reaches 6.645 for class III obesity [2].

These findings are further supported by a large prospective cohort study involving 74,317 Swedish adults, which demonstrated a dose-dependent relationship between both body mass index (BMI) and waist circumference (WC) and the risk of VTE [134]. Importantly, the study identified WC—a marker of abdominal obesity—as a potentially superior predictor of VTE risk compared to BMI. A substantially elevated WC was associated with a 53% increased risk of VTE, even among individuals with a normal BMI. The study estimated that 23.7% of VTE cases could be prevented by normalizing WC, compared to 12.4% by normalizing BMI to below 25 kg/m^2^.

In addition to increasing the overall risk of VTE, obesity significantly influences its clinical presentation, particularly in postoperative patients. One notable trend is the shift in the manifestation of VTE toward PE rather than isolated DVT as BMI increases. Specifically, the likelihood of a VTE presenting as a PE is 58.5% in patients with obesity, compared to 54% in overweight individuals and 49.4% in those with normal weight. This pattern is further reinforced when considering absolute body weight. Among patients who developed VTE, 67% of those weighing over 100 kg experienced a PE, in contrast to 53% of individuals weighing between 50 and 100 kg. Additionally, higher BMI is associated with an earlier onset of VTE following surgery, suggesting that obesity may accelerate thrombotic complications in the postoperative period [15].

Mechanistically, the elevated risk of VTE in obesity is largely attributed to a pro-thrombotic state driven by chronic low-grade inflammation and impaired fibrinolysis. Individuals with obesity exhibit significantly higher levels of fibrinogen, a response primarily induced by pro-inflammatory signals originating from AT [141].

Research has shown that adipocytes themselves can directly contribute to the coagulation cascade. In murine models, adipocytes have been identified as a source of coagulation factor VII, particularly under obese conditions. Complementary findings in humans highlight the role of factor VIII (FVIII), revealing a strong, dose-dependent relationship between elevated FVIII levels and the risk of recurrent venous thrombosis—even within overweight subgroups [142,143].

These findings underscore FVIII’s utility as a biomarker for improving thrombosis risk prediction and reinforce the close association between excess body weight, coagulation activity, and thrombotic outcomes. Another hallmark of the pro-thrombotic state in obesity is the elevated level of PAI-1, a key inhibitor of fibrinolysis. This increase is particularly pronounced in individuals with high volumes of VAT [144].

The risk of recurrent VTE is significantly elevated in individuals with MetS—a condition defined by the co-occurrence of central obesity, hypertension, insulin resistance, and dyslipidemia [145]. The recurrence rate of VTE increases in a stepwise fashion with each additional component of metabolic syndrome, ranging from 6.8% in individuals with none of the components to 37.1% in those exhibiting all four [16].

Further compounding this risk, patients frequently experience substantial weight gain following an acute DVT. On average, individuals gain 7.12% of their body weight within six months post-DVT. This effect is more pronounced among hospitalized patients, who gain an average of 8.6%, compared to 4.9% in outpatients [128].

Interestingly, in a study involving cancer patients with PE, a lower visceral fat thickness (VFT) was paradoxically associated with higher 90-day mortality. Survivors exhibited a median VFT of 130 mm, while non-survivors had a significantly lower median VFT of 85.8 mm. These findings suggest that, in the context of cancer, reduced visceral fat may serve as a surrogate marker for cachexia and metabolic deterioration—both of which are linked to poorer clinical outcomes [140].

## 6. Conclusions and Future Perspective

Venous thromboembolism results from a multifactorial interplay of various risk factors, with obesity recognized as a well-established independent contributor. This association is supported by both population- and hospital-based studies. Of note, obesity is characterized by a chronic low-grade inflammatory state, as evidenced by elevated C-reactive protein and increased levels of pro-inflammatory cytokines such as IL-6, IL-8, and TNF-α, all of which may foster a prothrombotic environment. Additionally, dysregulated adipokine secretion further disrupts the coagulation balance.

Given the complex etiology of VTE, various risk assessment models have been developed to inform prophylactic strategies. Obesity is acknowledged as a significant risk factor in many of these tools, including the Khorana score for cancer-associated thrombosis and other models used for patients hospitalized with acute medical conditions or undergoing major surgeries such as total hip arthroplasty.

Despite this recognition, obesity remains underrepresented in several widely used risk stratification tools, potentially leading to an underestimation of VTE risk and related complications in this population. Anticoagulation therapy in obese patients also presents unique challenges. According to European Heart Rhythm Association (EHRA) guidelines, direct oral anticoagulants (DOACs) are not routinely recommended for individuals with a BMI greater than 40 kg/m^2^ due to limited data on efficacy and safety. In such cases, therapeutic drug monitoring—such as measuring plasma drug concentrations or anti-Xa activity—is advised to ensure adequate anticoagulation [146].

Moreover, recent evidence highlights the potential VTE-protective effects of glucagon-like peptide-1 receptor agonists (GLP1-RAs), agents commonly used in the treatment of T2D and obesity. In a 2025 study, Chiang et al. used a target trial emulation approach involving over 540,000 matched patients from a global health database to compare GLP1-RAs with dipeptidyl peptidase-4 inhibitors (DPP4is) over a 12-month period [147]. The results demonstrated a significantly lower risk of VTE among GLP1-RA users (hazard ratio 0.78), including reduced risks of pulmonary embolism and deep vein thrombosis. These protective effects were consistent across BMI categories, age groups, and sexes, and were also evident when compared to other classes of diabetes medications.

The authors suggest that both the weight-reducing effects and potential direct antithrombotic properties of GLP1-RAs may contribute to the observed reduction in VTE risk. While the study’s large sample size and rigorous methodology strengthen its conclusions, limitations remain—notably the reliance on diagnostic coding, the absence of detailed weight-loss data, and the potential for residual confounding. Nevertheless, these findings add to the expanding body of evidence supporting the metabolic, cardiovascular, and now thromboembolic benefits of GLP1-RAs in patients with T2DM. Future research should aim to confirm these results in broader, more diverse populations and further investigate the biological mechanisms underlying the potential antithrombotic effects of GLP1-RAs.

## Figures and Tables

**Figure 1 ijms-26-10292-f001:**
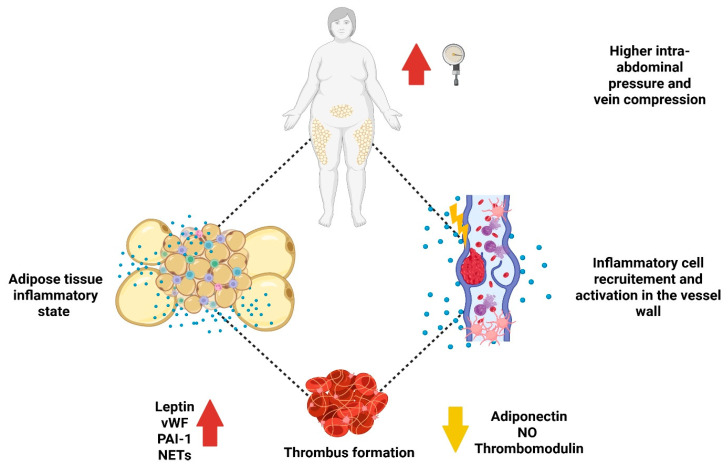
This figure presents postulated interaction between obesity-“adiposopathy”, pro-inflammatory state and exaggerated thrombus formation. The figure was created in BioRender. Imiela, A.M.

**Table 1 ijms-26-10292-t001:** Table summarizes selected studies evaluating the role of leptin in obesity-mediated venous thromboembolism.

Author	Country	Year	Study Design	Total Patients	Main Conclusions
Ageno W. et al. [128]	Italy	2003	prospective observational study	72	A substantial increase in BMI is observed in patients during the 6 months subsequent to a VTE. Weight loss is crucial to reduce the risk of future thrombotic episodes.
Barba R. et al. [129]	Spain	2005	observational study	8845	In VTE patients, body weight (>100 kg) was not linked to an increased risk of recurrence or major bleeding compared to patients weighing 50–100 kg. BMI < 50 kg was related with higher risk of bleeding complications.
Horvei L.D. et al. [130]	Norway	2014	prospective, population-based cohort study	6379	The increased risk of VTE associated with obesity is linked to overall body fat.
Horvei L.D. et al. [131]	Norway	2016	prospective, population-based cohort study	17,802	An increase of over 7.5 kg was linked to a nearly 2-fold greater risk, and this risk amplification was most prominent in the cohort with baseline BMI ≥ 30.
Samuels J.M. et al. [132]	USA	2019	retrospective study	687	Among severely injured individuals, obesity is associated with a prothrombotic state manifesting as increased clot stability and impaired fibrinolytic activity.
Glise Sandblad K. et al. [133]	Sweden	2020	prospective cohort study	1,639,838	Late adolescent obesity is a risk factor for the development of VTE in adulthood. Compared to leaner peers, young men with BMI ≥ 30 have a nearly 3-fold increased risk of VTE. This risk increases to almost 5-fold for subjects with BMI ≥ 35.
Stewart L.K. & Kline J.A. [16]	USA	2020	retrospective study	151,054	MetS and its constituent components, including obesity, are significant risk factors for recurrent VTE.
Yuan S. et al. [134]	Sweden	2021	prospective cohort study	74,317	WC is the superior to BMI as VTE predictor. An elevated WC accounts for an estimated 23.7% of VTE cases—nearly double the 12.4% attributed to BMI.
Ten Cate V. et al. [12]	Germany	2021	prospective cohort study	693	Obesity was linked to a 50% lower risk of recurrent VTE or death, but leptin resistance negated its protective role, ruling it out as the cause of this paradox.
Frischmuth T. et al. [135]	Norway	2022	population-based case–cohort study	1470	The concurrent presence of obesity and multiple prothrombotic genetic variants additively increases the overall risk of VTE.
Druar N.M. et al. [136]	USA	2022	retrospective study	1,002,831	In hospitalized patients with UECVCs, obesity is a significant predictor for both UEDVT and PE.
Sari M. et al. [137]	Turkey	2022	cross-sectional, retrospective, single-center study	582	A significant association exists between obesity and thrombosis risk in individuals with cancer. Increased body weight, BMI ≥ 35, and waist circumference (97.58 cm for the medium-high risk group compared to 93.40 cm for the low-risk group) are key predictors of a higher risk score.
Mahmoud A. et al. [138]	Sweden	2023	prospective cohort study	1,068,040	There is a direct, dose–response relationship between early-pregnancy BMI and the long-term risk of post-pregnancy VTE. This elevated risk is present even in women with a high-normal BMI of 22.5–25 and is more than tripled for those with BMI ≥ 35 when compared to lean women.
Frischmuth T. et al. [139]	Norway	2024	prospective, population-based cohort study	36,341	Excess body weight, defined as a BMI ≥ 25, accounts for nearly 25% of the total incidence of VTE at the population level.
Taşçı F. et al. [140]	Turkey	2025	retrospective study	75	For patients with cancer-associated PE, a decreased visceral fat thickness (VFT) was significantly correlated with an increased risk of 90-day mortality. This association may be linked to altered AT catabolism and remodeling that occurs during malignancy, where diminished fat reserves could signify greater disease severity and progression.
Santoyo Villalba J. et al. [15]	International (based on the RIETE registry, which includes data from 204 hospitals across 25 countries)	2025	observational study	3196	Among patients who have undergone surgery, higher BMI and weight correlate with an increased probability of developing a PE as opposed to an isolated DVT. Furthermore, the onset of VTE tends to occur more rapidly after surgery in individuals with obesity.

Abbreviations: AT–adipose tissue; BMI–body mass index; DVT–deep vein thrombosis; MetS–metabolic syndrome; PE–pulmonary embolism; UECVC–upper extremity central venous catheters; UEDVT–upper extremity deep vein thrombosis; VFT–visceral fat thickness; VTE–venous thromboembolism; WC–waist circumference.

## Data Availability

No new data were created or analyzed in this study. Data sharing is not applicable for this article.

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
