# Peer review of "The Interplay Between Obesity and Venous Thromboembolism: From Molecular Aspects to Clinical Issue"

_ijms, 2025, doi:10.3390/ijms262110292_

Round 1

Reviewer 1 Report

Comments and Suggestions for Authors

Recommendation: Reconsider after major revisions

Comments:

In this manuscript, the review article, on which the title: The interplay between obesity and venous thromboembolism: from molecular aspects to clinical issue could be reconsider,but after major revisions as followings:

  1. Firstly, The interplay between obesity and venous thromboembolism: from molecular aspects to clinical issueshould also be summarized with graphic abstract in clearly and simplified words. Suggestions to be added the clearly graphic abstract again, if possible.
  2. If possible, the Endocrine functionsincluding the relationship between leptin and Adiponectin should also be described and summarized in clearly and simplified words. Suggestions to be added the clearly graphic abstract again.
  • And additionally, in section 3: “Inflammation”section, the relationship between triggers of obesity-induced inflammation and  Macrophages should also be described and summarized in clearly and simplified words. Suggestions to be added the clearly graphic abstract again, if possible.
  • Lastly, Leptin and clinical implicationsshould also be summarized  in clearly and simplified words. Suggestions to be added the clearly graphic abstract again, if possible.

Author Response

In this manuscript, the review article, on which the title: “ The interplay between obesity and venous thromboembolism: from molecular aspects to clinical issue ” could be reconsider, but after major revisions as followings:

Query 1: Firstly, The interplay between obesity and venous thromboembolism: from molecular aspects to clinical issueshould also be summarized with graphic abstract in clearly and simplified words. Suggestions to be added the clearly graphic abstract again, if possible.

Answer 1: As requested, we have added a graphical abstract to clearly illustrate the interplay between obesity and venous thromboembolism.

Q2: If possible, the Endocrine functionsincluding the relationship between leptin and Adiponectin should also be described and summarized in clearly and simplified words. Suggestions to be added the clearly graphic abstract again.

Answer 2: We are enormously grateful for this valuable suggestion. We have added the following:

This inflammation is driven by stress within hypertrophied adipocytes, which leads to localized hypoxia, cellular dysfunction, and ultimately, cell death.

Recent research pinpoints adipocyte death as a key driver of this inflammation, as their large size prevents efficient, anti-inflammatory removal by single macrophages [11]. Instead, this size issue triggers a pro-inflammatory response where multiple macrophages surround the dead cell to form characteristic crown-like structures (CLS). The primary cause of this inflammation is a functional change in the tissue’s existing macrophages, rather than by the recruitment of new monocytes from the blood [11].

Q3: “Inflammation”section, the relationship between triggers of obesity-induced inflammation and  Macrophages should also be described and summarized in clearly and simplified words. Suggestions to be added the clearly graphic abstract again, if possible. Leptin and clinical implicationsshould also be summarized  in clearly and simplified words. Suggestions to be added the clearly graphic abstract again, if possible.

Answer 3: Thank you for this suggestion. We have modified a paragraph as following:

This dysregulation is particularly evident in the opposing actions of leptin and adiponectin. Leptin promotes a pro-thrombotic state by impairing fibrinolysis and enhancing platelet activation, an effect supported by genetic studies linking its high levels to a significantly increased VTE risk [5]. In contrast, adiponectin is considered a potential protective factor, and its reduced levels are associated with worse outcomes, such as post-thrombotic syndrome [5,6]

Reviewer 2 Report

Comments and Suggestions for Authors

In this review paper, the authors review the role of obesity as an independent risk factor for venous thromboembolism (VTE), including deep vein thrombosis and pulmonary embolism, with risk rising in proportion to body mass index and abdominal adiposity. Dysfunctional adipose tissue fosters a pro-thrombotic environment through chronic low-grade inflammation, immune cell infiltration, and impaired fibrinolysis. Adipokines, particularly leptin, illustrate the complex obesity–VTE relationship, while the “obesity paradox” highlights the unexpected association between low leptin and worse outcomes in acute pulmonary embolism. Metabolic syndrome further amplifies recurrent VTE risk, underscoring the need to clarify molecular pathways and develop targeted preventive strategies.

The paper is extremely well written and thorough. I have no major criticisms.

Minor correction:

  1. Line 510: “Not” should be “note.”

Author Response

Query 1: Line 510: “Not” should be “note.”

Answer 1: We thank the reviewer for their positive assessment and careful reading of our manuscript. We have corrected the typographical error on line 510 from “not” to “note” as suggested.

Reviewer 3 Report

Comments and Suggestions for Authors

This is a well-structured, comprehensive, and timely review that systematically explores the complex relationship between obesity and venous thromboembolism (VTE). The authors successfully integrate molecular mechanisms with clinical data, making the review relevant to both researchers and clinicians.

I have raised some crucial comments:

  • Platelets play a multifaceted role in the pathogenesis of venous thromboembolism. Obesity profoundly alters platelet function through a multifaceted interplay of metabolic, inflammatory, and signaling pathways, contributing to thrombotic risk. The authors need to incorporate relevant content on this topic in the manuscript.
  • Stasis of blood flow (or venous stasis) is a critical and well-established factor in the development of venous thromboembolism. Obesity has a significant and predominantly negative impact on venous blood flow. the authors need to incorporate relevant content on this topic in the manuscript.

Author Response

This is a well-structured, comprehensive, and timely review that systematically explores the complex relationship between obesity and venous thromboembolism (VTE). The authors successfully integrate molecular mechanisms with clinical data, making the review relevant to both researchers and clinicians.

I have raised some crucial comments:

Query 1: Platelets play a multifaceted role in the pathogenesis of venous thromboembolism. Obesity profoundly alters platelet function through a multifaceted interplay of metabolic, inflammatory, and signaling pathways, contributing to thrombotic risk. The authors need to incorporate relevant content on this topic in the manuscript.

Answer 1: We are very grateful for this suggestion. We have modified a paragraph as following:

Furthermore, adiposity exacerbates platelet activity, leading to overly responsive platelets that are a core reason for the overall risk for thrombosis. This is demonstrated by the increased activation and aggregability observed in the platelets of obese individuals [7].

Query 2: Stasis of blood flow (or venous stasis) is a critical and well-established factor in the development of venous thromboembolism. Obesity has a significant and predominantly negative impact on venous blood flow. the authors need to incorporate relevant content on this topic in the manuscript.

Answer 2: We are enormously grateful for this valuable suggestion. We have added the following:

Obesity contributes to venous stasis through a clear mechanism known as obesity-induced caval vein compression syndrome [9]. The increased intra-abdominal pressure resulting from a high BMI and intra-abdominal fat accumulation is linked to a corresponding increase in the pressure differential observed across the diaphragm between the thoracic and abdominal vena cava segments. The resulting compression of the inferior vena cava hinders venous return, leading to reduced venous blood flow velocity and an increased risk for venous thrombosis [9].

Round 2

Reviewer 1 Report

Comments and Suggestions for Authors

Recommendation:  Accept in present form

Comments:

In this manuscript, the review article,on which the title :The interplay between obesity and venous thromboembolism: 2 from molecular aspects to clinical issue  could be acceptable in present form, and no needed to be revised further.

Reviewer 3 Report

Comments and Suggestions for Authors

The author has answered all the questions, and it is recommended to accept. Additionally, I have a suggestion for the author to write a review describing how obesity affects deep vein thrombosis by influencing platelet function.